



# Strategies for adapting to hazards and environmental inequalities in coastal urban areas: what kind of resilience for these territories?

Nathalie Long[1], Pierre Cornut [2], Virginia Kolb[1]

[1]LIENSs, La Rochelle université - CNRS, 17000 La Rochelle, France

[2]Faculté d'architecture et d'urbanisme, Université de Mons, Mons, Belgium

*Correspondence to:* Nathalie Long (nathalie.long@univ-lr.fr)

## Abstract

The ongoing phenomenon of climate change is leading to an upsurge in the number of extreme events. Territories must adapt to these modifications in order to protect their populations and the properties present in coastal areas. The adaptation of coastal areas also aims to make them more resilient to future events. In this article, we examine two strategies for adapting to coastal risks: holding the coastal line through hard constructions such as seawalls or ripraps and the managed retreat of activities and populations to a part of the territory not exposed to hazards. In France, these approaches are financed by a solidarity insurance system at the national level as well as local taxes. These solidarity systems aim to compensate the affected populations and finance implementation of the strategies chosen by local authorities. However, the French mainland coast generally attracts affluent residents, the price of land being higher than inland. This situation induces the presence of inequalities in these territories, inequalities which can be maintained or reinforced in the short and medium term when a defence strategy based on hard constructions is implemented. In such a trajectory, it appears that these territories would be less resilient in the long term, because of the maintenance costs of the structures and the uncertainties relating to the hazards (submersion, rising sea levels, erosion). Conversely, with a managed retreat strategy, inequalities would instead be done away with, since property and populations would no longer be exposed to hazards, which would cost society less and would lead these territories towards greater resilience in the long term. Only one social group would be strongly impacted by this strategy in the short term when they are subjected to a managed retreat to another part of the territory.

## 1 Introduction, state of the art and objectives

Coastal territories are nowadays areas with high stakes, on both the social and economic levels. These attractive territories host most of the world's major megalopolises with population densities higher than those of inland towns (Neumann et al., 2015). However, in a context of climate change, these same territories find themselves exposed to meteorological and





30 marine hazards, such as marine submersions, coastal erosion or rising sea levels. Urban growth must therefore adapt to these new environmental conditions and respond to issues of sustainability, resilience and equity between different social groups (Hurlimann et al., 2014). Three main types of adaptation strategy are being implemented today with different social and environmental impacts (Williams et al., 2018): holding the coastal line through the construction of hard protective structures, the managed retreat of properties and infrastructure, or mitigation. The choice between either strategy

35 is most often based on a cost/benefit approach, and therefore depends on the value of assets and properties present in the territory (Cooper and McKenna, 2008; André et al, 2016). This approach favours a certain equity because it is based on economic metrics that allow monetary cost comparisons; on the other hand, it excludes all human and social considerations, such as the capacity to adapt at the individual level, attachment to a place and memory of the risk, as well as the maintenance or reinforcement of inequalities (Boda, 2017; Füssel and Klein, 2006; Ramm et al, 2018).

40 This approach can also favour or disadvantage certain social groups. Many studies have shown that environmental policies have different impacts according to social groups, generally disadvantaging the poorest, the most exposed to coastal risks and the most vulnerable from a social point of view (Wallace, 2012; Velez et al., 2018). Social justice, defined as a fair distribution of costs and benefits over the entire population, can be questioned here and supplemented by an environmental dimension, namely ecosystem protection (Cooper and McKenna, 2008; Dobson, 1999). In the same way, Eriksen et al.

45 (2011) specify that an adaptation to climate change can only be deemed successful if it achieves social and environmental sustainability by factoring in both social justice and environmental integrity. Thus, the exposure of the poorest populations to climate change is often used as a pretext for implementing adaptation measures, but they are rarely evaluated according to this criterion and will not, in any case, alleviate either poverty or inequalities (Eriksen et al., 2011).

Coastal urban territories must therefore be studied as a system for taking into account both social and environmental

50 dimensions as well as their interrelationships. Understood as a system, the coastline can then be approached according to the theory of resilience as an evolving system that is able to adapt to changes, in this case concerning the climate. According to Holling et al. (2002), every system is in a dynamic of cycles which evolve at different spatial and temporal scales through different phases represented by the well-known "figure eight" (Holling, 2001). By adopting the key principles of the theory of resilience (Holling and Gunderson, 2002) and, with regard to the characteristics of coastal urban

55 territories, the observable changes are effectively, either continuous or episodic, uniform or highly variable spatially, destabilising or conversely a source of stability, which leads us to conclude that management policies must be tailored to these spaces, more with a view to ensuring the resilience than the stability of these spaces, endeavouring to maintain them in their current state (Walker et al., 2004; Curtin and Parker, 2014). They must allow the development of new trajectories, while accepting a significant degree of uncertainty as to the future development of the system (Redman and Kinzig, 2003).

60 In this article, we will examine the impacts in terms of the inequalities of two coastal risk management policies: holding the coastal line by building seawalls or ripraps, and managed retreat. Broaching the matter through environmental inequalities (EI) makes it possible to tackle, in parallel, social and environmental aspects as described previously by Eriksen



et al. (2011), which the concept of social justice does not allow. In general terms, environmental inequalities are defined as intra- and inter-generational social inequalities partly determined by the state of the environment and partly by the way

society is organised. More specifically on the coast, they arise when a social group is disproportionately affected by a risk compared to other social groups (Pye et al., 2008; Deldreve, 2015; Brulle and Pellow, 2006). Brulle and Pellow (2006) also add that environmental inequalities are products of society and its dynamics and, in coastal areas, result from the particular way in which this society is organised. Inequalities are thus defined in relation to others or to a benchmark which fully justifies their use to analyse the impacts of coastal management strategies on the populations concerned and

their living area. Inequalities also reflect a greater or lesser distance between the different social groups (Uslaner and Brown, 2005) and can compromise the expression of social solidarity (Durkheim, 1964), which, in crisis and post-crisis situations, as during a natural disaster, can be problematic for finding a new state of equilibrium. Coastal risk management policies must not compromise this ability of territories to be resilient, in a way which is both human (less inequality for more solidarity and therefore sustainability of the social system) and environmental (protection of the environment and

of its role as a buffer zone in the face of marine hazards).

In coastal urban areas, there are many inequalities (Kolb et al., 2014). Only 3 types of EI will be discussed in this article because they are intrinsically linked to the management of coastal risks: inequalities in access to land, inequalities in exposure to risk and inequalities in access to the coastline perceived as an amenity (see appendix A for the definition of each of these inequalities). Territorial inequalities through economic development and infrastructural services can also be

present in coastal areas but are not directly linked to risk management, etc. In the current context of increasing hazards, the question posed is whether the strategies for adapting to coastal risks more particularly will exacerbate or, on the contrary, alleviate the environmental inequalities already present in these territories and so influence their resilience? We propose to engage in this examination on two timescales: in the short term, through one-off actions carried out, and in the long term, taking into account their sustainability for future generations. In other words, are environmental changes and

the societal response to them likely to increase or decrease environmental inequalities? The case of the northern part of Charente-Maritime (France) will be taken as an example, even if the scope of this argument is intended to be more general. In the second part, we will present the French context by describing the coastal risk management strategies and their mode of financing so as to establish the framework for our examination in order to engage with the methodological explanation in the third part. The results will be presented in the fourth part and then discussed in the fifth part before concluding this

article.



## 2 The French context through the La Rochelle case study

### 2.1 Presentation of the study area

During the INEGALITTO project, we had the opportunity to study a coastal territory in the West of France, around La Rochelle. This territory includes the coastal municipalities and their neighbouring municipalities in the agglomeration of

La Rochelle, to which the municipality of Charron has been added to the north, in order to preserve a certain geographical continuity on the scale of Pertuis Charentais (25 municipalities studied, figure 1). Located on the Atlantic coast, this area is fairly urbanised and attractive with two seaside resorts, namely La Rochelle and Châtelaillon-Plage. La Rochelle is the main city with nearly 76,000 inhabitants. It also hosts the majority of jobs, infrastructures and amenities. The population density in the agglomeration of La Rochelle is the highest in the department with 415 inhabitants/sq.km but features a

very high spatial variability, with 50% of the population present in the city centre. The coastal zone is somewhat charac- terised by an ageing population while the inland municipalities are fairly dynamic thanks to the arrival of families with young children. From a morphological and sedimentary point of view, the coast is firstly silty to the north in the Baie de l'Aiguillon, then featuring cliffs north of La Rochelle, followed by a few sandy beaches, artificially maintained by regular refilling with sand for some, or beaches with pebbles or silt once more, through to the south of the study area.

Charron, the second specific study site within the framework of the INEGALITTO project, is located north of the Pertuis Breton. This municipality is characterised by a productive economy, based on agriculture and shellfish farming. It has fewer than 2,000 inhabitants but occupies a strategic position between two municipalities with employment areas, La Rochelle in the south and Luçon in the north, in the department of La Vendée. Here, the coast is fairly silty making it less attractive for tourism, but more natural.

This study area was strongly impacted during Storm Xynthia in February 2010. Material damage was significant and lives were lost. Following this event, the legislation and its implementation were strengthened at the national level. The INE- GALITTO research project made it possible to question the management of coastal risks in France, over the decade following this major event.

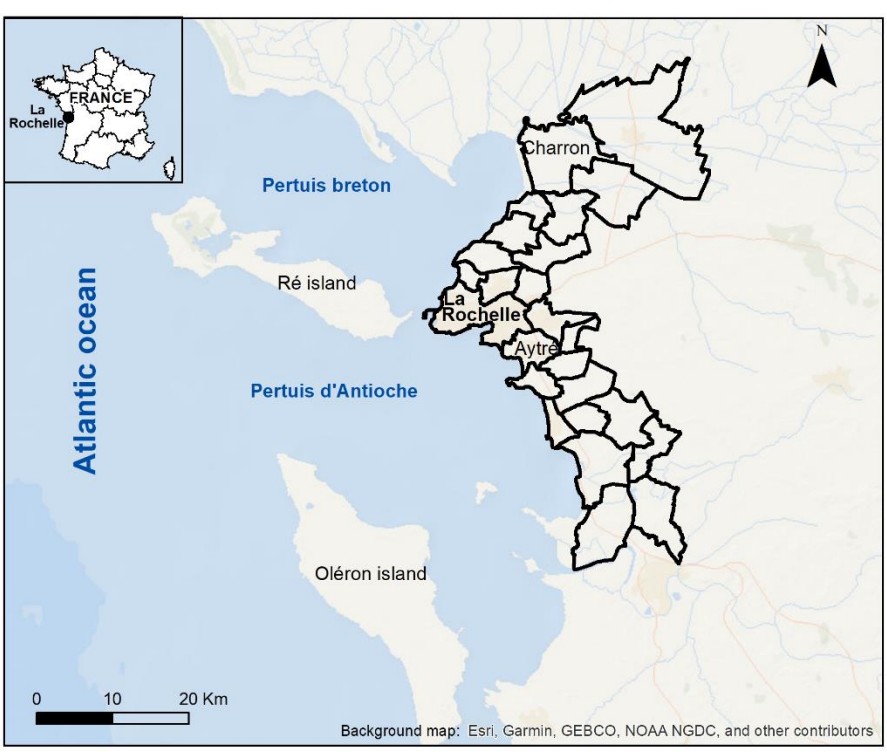

**Figure 1: The 25 coastal municipalities studied close to La Rochelle, the city centre, on Atlantic coast (the black lines represent the municipal boundaries, Aytré – Charron are the main studied municipalities, source: IGN BDTopo®).**

**2.2 Coastal protection strategies and funding mechanisms at national and local level**

Faced with natural disasters, the French state has introduced original compensation schemes for damage suffered following a natural disaster, based on solidarity at the national level first, then local since 2018. In terms of protection and prevention, three main strategies have been initiated in France and have different environmental and societal impacts (Williams et al., 2018). We are only covering two of these three strategies here :

(i) Holding the coastal line by constructing hard defensive works such as seawalls or ripraps. This strategy makes it possible to protect the properties directly exposed to hazards and to keep them in place. These properties have a value that is greater than the cost of building a seawall. This strategy favours maintaining the territory and its population in its current structure and organisation. It can, however, have a significant negative impact on the environment itself and its amenities: for example, for certain types of sandy coastline, the disappearance of the beach in the short and medium term;

(ii) The managed retreat of the properties. This time, the assets have a lower value and it is then decided to demolish them rather than keep them near the coast in an area exposed to hazards. This strategy leads to a spatial reconfiguration of the



territories and a displacement of the population. The positive impact on the environment is also important with a return to the naturalness of the coast. Portions of territory can thus be returned to the sea and play their role of buffer zone during storms. However, for the populations concerned, it can be more overwhelming from an emotional and organisational point of view, depending on the capabilities of the individuals (Sen, 1997).

To set up these risk management strategies, the so-called "Barnier Law" in 1995 introduced various risk prevention plans,
including PPRLs (coastal risk prevention plans) as of 1997 (and updated in 2011 following Storm Xynthia and the exceptional floods in the Var department, France). These plans aim to reduce the vulnerability of people and property by defining compulsory prevention, protection and safeguard measures and by establishing zoning to control construction in exposed areas. To implement these comprehensive measures at the local level, PAPIs (flood prevention and action programmes) have been defined. Finally, to finance these actions, the Fund for the Prevention of Major Natural Risks, known
as the Barnier Fund, was also created by this law in 1995. This fund also makes it possible to finance or co-finance i) expropriations or amicable acquisitions of property following a disaster and when the threat is still present, ii) studies and construction work leading to a reduction in vulnerability or the compliance of structures, and iii) information campaigns. The Barnier Fund is financed by a compulsory levy on the additional premium paid by all policyholders under the coverage against natural disasters: 12% on home insurance contracts and 6% on vehicle insurance contracts. This represents
around 200 million euros paid per year by insurance firms. This procedure, commonly called the Cat-NAT system (short for natural disasters in French), is based on national solidarity because the financing is organised on the basis of an additional premium paid by all holders of comprehensive insurance policies. But in the wake of the MAPTAM law of 2014, relating to the modernisation of territorial public action and the affirmation of metropolises, authority for GEMAPI (aquatic environment management and flood prevention) has since 1 January 2018 been assigned to municipalities and
their EPCIs (public inter-municipal cooperation establishments). Although the State, regions and departments were previously the main contracting authorities and co-financers of defence infrastructures, all of this is now managed at the local level. This decentralisation of authority allows the EPCIs to levy a new tax, known as the GEMAPI tax, varying from one local authority to another and capped at €40/inhabitant. It is therefore also a solidarity mechanism, but this time on a local scale (municipal and inter-municipal), which was recently put in place.

**3. Data and methods**

**3.1. Results from the INEGALITTO project**

The INEGALITTO project questioned the environmental inequalities produced by risk management policies in coastal urban areas. The chosen approach was mixed and based on both a quantitative and a qualitative study. The first part of this project consisted in mapping environmental inequalities using indicators to define inequalities in access to natural



and anthropogenic amenities, inequalities in exposure to natural and industrial risks, and inequalities on the economic
level. These inequalities were then compared with social ones, defined by socio-demographic data at the household level.
Starting from the hypothesis that the most socially vulnerable populations are the most exposed to risks and have the least
access to natural amenities, our results have shown that in the case of the La Rochelle agglomeration, these two hypotheses
do not hold true, in that the better-off populations may also be exposed to risks and find themselves some distance from

natural amenities (Long et al., 2019). The explanation lies in the socio-spatial structure of the city of La Rochelle, which
is superimposed on the coastal effect in the distribution of households over the whole of the agglomeration.

The second part of the project favoured a qualitative approach, through interviews with local politicians, associations and
residents. The commune of Aytré, to the south of La Rochelle, as well as the commune of Charron, to the north of the
study area, were chosen to carry out these surveys. These two municipalities have benefited from the construction of

seawalls to protect the assets and populations from future marine submersions but have also experienced managed retreat
with the demolition of many residential houses, and so the disappearance of part of the neighbourhoods impacted by the
flooding during Storm Xynthia. According to the surveys carried out, in Aytré, the population questioned seems to have
grasped the fact that living near the coast remains a privilege and that the downside is being exposed to these risks. This
risk has become "banal" in the sense that it is factored into their everyday life. The population does not deny it but tucks

it away as an afterthought. The populations furthest from the coast in this town are however beginning to question why
they have to pay (via the insurance premium and the GEMAPI tax) to protect those who persist in living in areas exposed
to coastal risks. In Charron, the population is still scarred by the disaster following Storm Xynthia. The spontaneous
solidarity which manifested itself immediately after the storm allowed the constitution of a still existing network of so-
ciability but today, part of this network is keen to move on and project a more positive image of the municipality. On the

other hand, the interviews unanimously reveal a sense of injustice when it comes to the treatment of territories, compared
to other municipalities where the procedures and the work to ensure protection by hard structures were implemented much
faster. Here it is inequality in the capacity to challenge public authorities that is often cited, as well as an inequality of
treatment between the territories.

**3.2. Method**

To assess the effects of the hazard, as well as the choice of defence strategies and financing mechanisms regarding envi-
ronmental inequalities, we have chosen to model the different situations as follows. According to the results of the quan-
titative study of the INEGALITTO project, it emerges that it is important to know the status of households (owner or
tenant) who live on the coastal strip, to assess the impact of an adaptation strategy, the properties that can be demolished

or maintained and protected. The consequences for the households concerned will then be of varying degrees of severity.
On the other hand, in terms of financing these strategies, the status of the household is irrelevant; as seen above, each
individual contributes indirectly (insurance premium and the GEMAPI tax) to the financing of these strategies. Based on





these criteria, in the French context, three social groups are proposed: two groups at the municipal level: private owners living on the coast (POs), other inhabitants of costal municipalities (ICMs) and a final group dubbed inhabitants of non-coastal municipalities (INCMs). Subsequently, two timeframes were considered: the short term, when defence structures are built or managed retreats carried out quickly after an event; and the long term, by taking into account future generations from the three population categories. In the latter case, our working hypothesis is an identical continuation of the coastal risk management policy and its mode of financing, whatever the evolution of the hazard. Finally, we apply an adaptation strategy to these 6 cases (3 groups * 2 timeframes) either by maintaining the coastline through the construction of hard defence structures, or through managed retreat, so obtaining 12 situations (Table 1).

| | Hold the coastal line (seawall) (HL) | | | Managed retreat (MR) | | |
|---|---|---|---|---|---|---|
| **Short term** | **Private owners (POs)** | **Other inhabitants of coastal municipalities (ICMs)** | **Inhabitants of non-coastal municipalities (INCMs)** | **Private Owners (POs)** | **Other inhabitants of coastal municipalities (ICMs)** | **Inhabitants of non-coastal municipalities (INCMs)** |
| **Long term** | **Private owners (POs)** | **Other inhabitants of coastal municipalities (ICMs)** | **Inhabitants of non-coastal municipalities (INCMs)** | **Private Owners (POs)** | **Other inhabitants of coastal municipalities (ICMs)** | **Inhabitants of non-coastal municipalities (INCMs)** |

Table 1: presentation of the 12 situations analysed according to the strategy, the timeframe and the population group

For each of these situations, we evaluated, on the one hand, the costs borne by the populations and, on the other, the advantages obtained and the drawbacks suffered by these populations. This evaluation is of a qualitative nature, aiming above all to differentiate the situations of the three social groups identified. The methodology for evaluating the costs incurred is simply based on the application of the French financing and insurance law. Regarding the advantages obtained and the drawbacks suffered, we have defined a series of qualitative indicators based on the literature concerning EIs (Kolb et al., 2014) and on our knowledge of the field and our expertise acquired within the INEGALITTO project. These indicators are defined in Appendix B. They are therefore classified according to the main types of environmental inequalities: economic or social access inequalities (Indicator: Economic and property values), access to environmental amenities (Indicators: Accessibility to the coast, Environmental evolution of the coast); risk exposure inequalities (Indicator: Natural hazard exposure); and finally social and cultural inequalities (Indicators: Inhabitant feeling, Sense of place, Social cohesion). These indicators are estimated for each of the 12 situations according to a simple qualitative scale: improvement or preservation (with a nuance depending on whether the populations are high concerned or low concerned), neutrality, degradation (with the same nuance). Knowledge of the field and the interviews we conducted enabled us to assess these indicators for each situation.





For greater clarity, the qualitative assessment scale of the indicators is represented in the graph by colours (Table 2). The boxes are coloured blue, white or burgundy depending on whether the indicator shows an advantage, a neutral situation or a drawback respectively. The intensity of the colour is in certain cases reduced to show that the advantage or the drawback is less important, given the geographical distance of the populations considered.

| Advantage (high concerned) | Advantage (low concerned) | Neutral | Drawback (low concerned) | Drawback (high concerned) |
|---|---|---|---|---|

**Table 2: Choice of colours for the qualitative representation of indicators**

## 4. Results

### 4.1 Costs incurred by the strategies

#### 4.1.1 Hold the coastal line strategy

In the *short term*, the cost of the seawall or riprap concerns its construction. It is financed by the Barnier Fund, which applies to all French insurance policyholders. Individually, the POs, ICMs and INCMs (Table 1) contribute to the financing at the same level via their insurance contract. Collectively, in terms of the solidarity mechanism, there is a double transfer of funding from the INCMs and the ICMs to the POs whose property is protected by the coastal defence (figure 2). They participate indirectly in financing the construction of the coastal defence, without being directly affected by this risk.

In the *long term*, the cost of the coastal defence boils down to its maintenance, financed by the GEMAPI. Only the POs and the ICMs still bear a cost (less than for construction), and there is a simple transfer of solidarity from the ICMs to the POs.

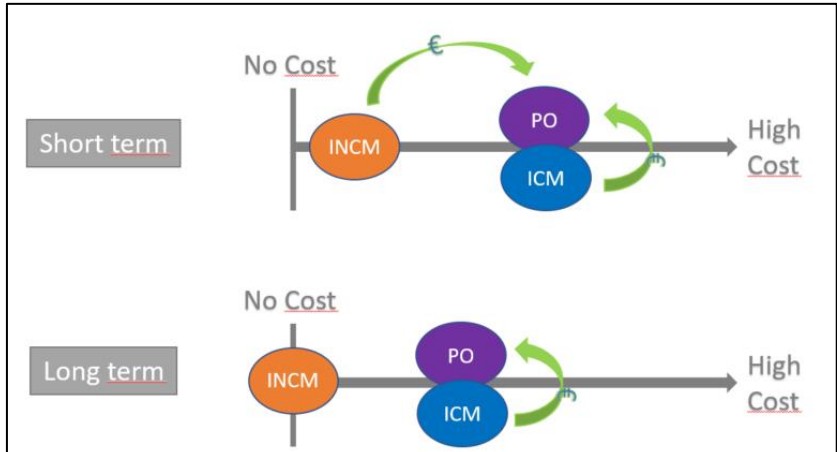

**Figure 2: Evolution of cost for the 'Hold the coastal line' strategy**





### 4.1.2 Managed retreat strategy

In the short term, the economic cost of demolition presents the same configuration as that of the coastal defence, both in terms of contribution levels and in solidarity transfers (figure 3). In the long term, however, the situation is radically different given the elimination of the assets: there are no more properties exposed because the POs have left; this cancels 240 out the costs for the INCMs and ICMs.

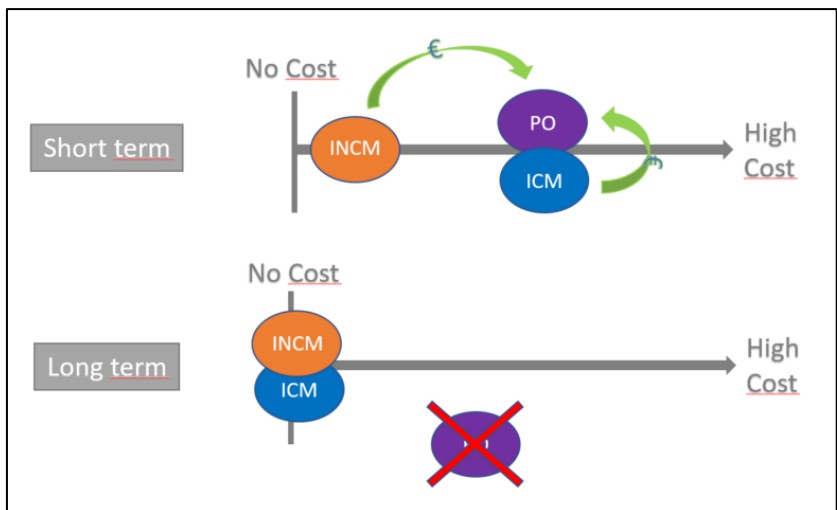

**Figure 3: Evolution of cost for the 'Managed retreat' strategy**

### 4.2 Advantages and drawbacks suffered

245 We compare the two strategies here in the form of a table, on the basis of the defined indicators. Interpreting the tables therefore makes it possible to easily summarise the advantages and drawbacks of the three population groups, for the two timeframes and the two adaptation strategies.

### 4.2.1 In the short term

According to Table 3, in the short term and according to a coastal maintenance strategy, for all the indicators, the POs 250 and ICMs see their advantages preserved, whether in terms of the value of their property or their well-being, among other factors. French citizens are barely concerned except for accessibility to the coast as a tourist destination. On the other hand, for the managed retreat strategy, the POs are somewhat disadvantaged with a loss of their property, their living space and being displaced away from the coast, although, on the other hand, they become less exposed to coastal risks, or even not at all. For the ICMs, only social type indicators indicate a loss, because these population movements cause a 255 loss of social cohesion for the affected territory. Economic indicators or those linked to coastal amenities are more positive





for these populations who can once again benefit from a coast that has become more natural again. Finally, we can hypothesise that the further we move away from the coastal regions, the less the populations are affected by the changes in these territories.

| Adaptation strategies and social groups / Which indicators | Hold the coastal line (seawall) | | | Managed retreat | | |
|---|---|---|---|---|---|---|
| | Private owners (POs) | Other inhabitants of coastal municipality (ICMs) | Inhabitants of non-coastal municipalities (INCMs) | Private owners (POs) | Other inhabitants of coastal municipality (ICMs) | Inhabitants of non-coastal municipalities (INCMs) |
| **Economic and property values** | Preserved | Preserved | Neutral | Loss of the real property but financial compensation | Preserved | Neutral |
| **Accessibility to the coast** | Preserved | Preserved | Preserved (low concern) | Decrease | Preserved | Preserved (low concern) |
| **Environmental evolution of the coast** | Art.* | Art.* | Art.* (low concern) | Renat.** | Renat.** | Renat.** (low concern) |
| **Natural hazard exposure** | Protection of buildings and inhabitants and lower exposure | Protection of territory and lower exposure | Protection of territory (low concern) | Decrease | Decrease | Neutral (no concern) |
| **Inhabitant feeling** | Well-being | Well-being | Neutral | Anxiety/ stress linked to relocation | Empathy | Empathy (low concern) |
| **Sense of place** | Preserved | Preserved | Neutral | Total loss | Decrease (Indirect concern) | Neutral |





| Social cohesion | Preserved | Preserved | Preserved (low concern) | Decrease | Decrease | Neutral |
|---|---|---|---|---|---|---|

**Table 3: Short term comparative advantages & drawbacks (*: Artificialization; **: Renaturation)**

### 4.2.3 In the long term

In the long term, the POs enjoy as many advantages as in the short term in the event of protection by coastal defences: their homes and living areas are maintained over time and they continue to benefit from their property while being protected from coastal risks. In the event of managed retreat, however, the POs no longer exist (the properties having been demolished) and the ICMs benefit from a coast which regains its status as a common good accessible to all (Table 4).

| Adaptation strategies and social groups / Which indicators | Hold the coastal line (seawall) | | | Managed retreat | | |
|---|---|---|---|---|---|---|
| | Private owners (POs) | Other inhabitants of coastal municipalities (ICMs) | Inhabitants of non-coastal municipalities (INCMs) | Private owners (POs) | Other inhabitants of coastal municipalities (ICMs) | Inhabitants of non-coastal municipalities (INCMs) |
| Economic and property values | Preserved | Preserved | Neutral | | Preserved | Neutral |
| Accessibility to the coast | Preserved | Preserved | Preserved (low concern) | | Preserved | Preserved (low concerned) |
| Environmental evolution of the coast | Art.* and possible loss of amenity | Art.* and possible loss of amenity | Art.* (low concerned) | | Renaturation | Renaturation (low concerned) |
| Natural hazard exposure | Protection of buildings and inhabitants and lower exposure | Protection of territory and lower exposure | Protection of territory (low concerned) | | Lower exposure | Neutral (no concerned) |
| Inhabitant | Well-being | Well-being | Neutral | | Well-being | Neutral |





| | | | | | | |
|---|---|---|---|---|---|---|
| **feeling** | | | | | | |
| **Sense of place** | Preserved | Preserved | Neutral | | Preserved | Neutral |
| **Social cohesion** | Preserved | Preserved | Preserved (low concerned) | | Preserved | Neutral |

**Table 4: Long-term comparative advantages & drawbacks (*: Artificialization)**

### 4.3 Summary of results

The summary of the results is presented by a schematic representation: each social group is positioned according to the
costs borne on the horizontal axis and according to the advantages/drawbacks in the vertical axis (figure 4). The main
conclusions that emerge from observation of the graphs are:

i. In the short term, the three population groups are still required to make a financial contribution by paying the
additional premium via their home and/or vehicle insurance contracts, which shores up the Barnier Fund. How-
ever, the consequences of these strategies are rather favourable to the POs and to a lesser extent to the ICMs in
the case of protection by a hard structure, but are largely unfavourable to the POs and rather favourable to the
ICMs (except on social aspects), in the case of a managed retreat.

ii. In the long term, however, as we made the assumption of no change in the insurance system, the additional
insurance premium remains compulsory, the three population groups thus continue to contribute indirectly to the
Barnier Fund. Only the GEMAPI tax is to be taken into consideration for the ICMs, assuming that the POs which
had to move have become either ICMs or INCMs. If there are no coastal defences to maintain and a reduction
in or even the elimination of assets to protect, we can then assume that the GEMAPI tax will be revised down-
wards by the EPCIs. The impacts are neutral or thereabouts for the INCMs and positive for the ICMs.

iii. Solidarity transfers to POs are present in the event of the construction of a hard coastal defence as in the event
of managed retreat, but they persist in the long term in the first case.

iv. In the short as in the long term, the advantages for the POs are much greater in the case of the construction of a
hard coastal defence than in the event of managed retreat.




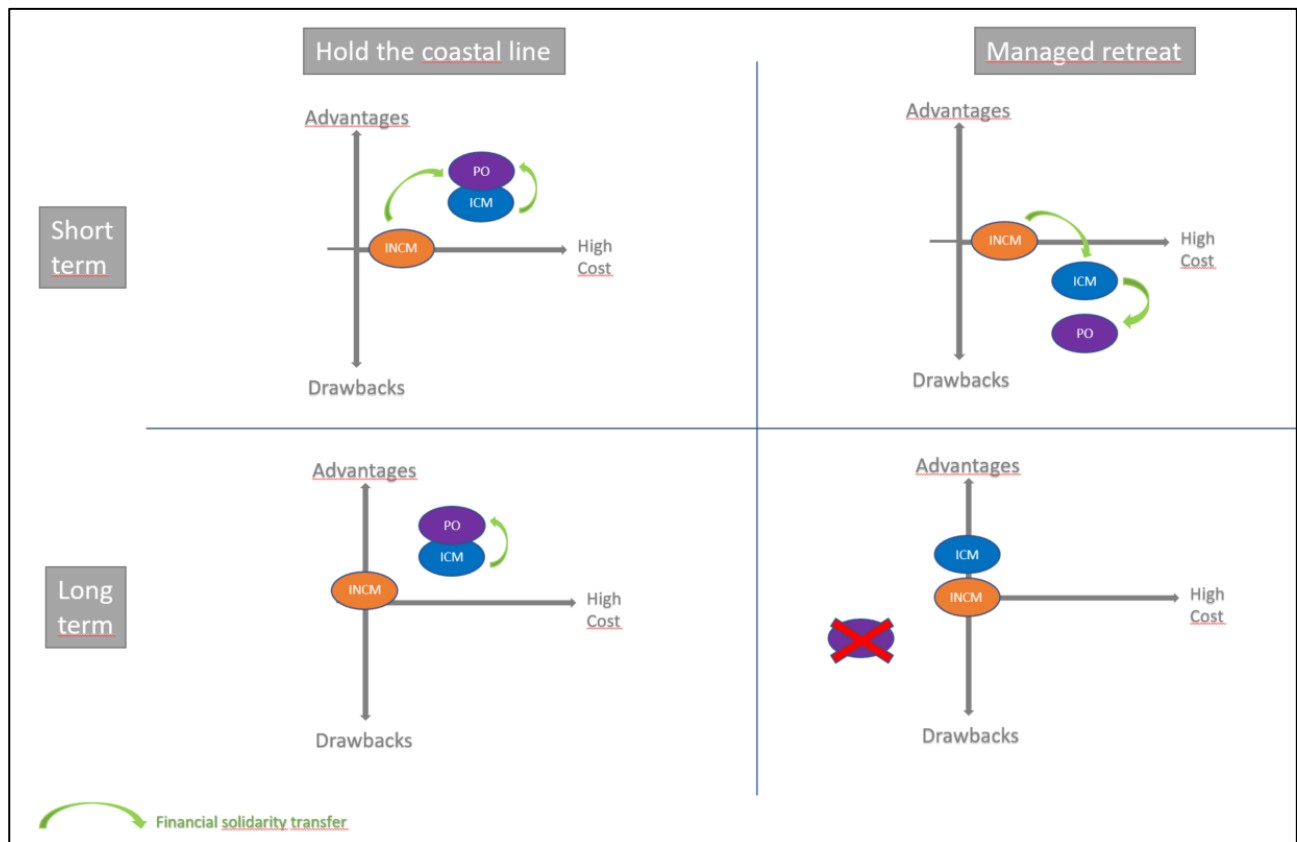

**Figure 4: Evolution of cost and comparative advantages and drawbacks, in the short and long term**

In terms of inequalities, we can deduce that inequalities in access to land are maintained on the coastal territory in the

case of the implementation of a protection strategy by the construction of a hard coastal defence. On the other hand, they

vanish in the case of a managed retreat following the demolition of these properties (table 5) and can possibly appear in

other parts of the territory. The spatial variability of land prices is in this case smoothed out over the short term as over

the long term according to a land-sea gradient.

The unequal access to the coast as an amenity in the event of the construction of a short-term coastal defence is either

maintained, if the structure is inaccessible (privatisation of the coast), or possibly reduced if the structure becomes a place

to walk. However, in the long term and even in the medium term, this amenity (especially if it is a beach) may end up

disappearing under the impact of the waves which will lead to erosion of the coast. In the case of a managed retreat

followed by renaturation of the coastline, this inequality should be eliminated, the coastline fully resuming its status as a

common good and the beach continuing to exist or evolving towards a naturally resilient system.





On the other hand, the inequality of exposure to risks is greatly reduced during the construction of a coastal defence. Even if a risk remains, it can again increase in the long term if the structure is not sufficiently maintained. It becomes non-existent in the case of a managed retreat because no human or structural assets are then exposed to the hazards.

Finally, the existing social inequalities are maintained when a coastal defence is built; moreover a managed retreat leads to a population movement which can destabilise a territory through a loss of social cohesion. However, in the long term,
a new social cohesion can also be built.

| Adaptation strategies and short/long term / Environmental inequalities | Hold the coastal line | | Managed retreat | |
|---|---|---|---|---|
| | **Short term** | **Long term** | **Short term** | **Long term** |
| **Unequal access to land** | Maintained | | Removed | |
| **Unequal access to the coast as an amenity** | Maintained or reduced | Reinforced | Removed | |
| **Unequal exposure to risks** | Scaled down | May increase again | Removed | |
| **Existing social inequalities** | Maintained | | Reinforced | Possibly reduced in a new managed retreat context |

**Table 5: Evolution of environmental inequalities**

**5. Conclusion and discussion**

The qualitative approach set out makes it possible to outline the effect of coastal risks and adaptation strategies on the EIs
of the territory considered. **Whatever the timeframe, owners of properties on the coast have a clear advantage in the case of protection by coastal defences, and these hard structures cost the local authority more than demolition, given the long-term maintenance required**. Do the strategies as well as their mode of financing by the Barnier Fund and then maintenance via the GEMAPI nevertheless reinforce the EIs? This would clearly be the case if there was an initial inequality between the owners of property on the coast and the rest of the inhabitants of France. In this specific
case, the owners of property on the coast, who may be labelled as a privileged population, benefit from protective measures by the construction of a coastal defence, financed by solidarity transfers, allowing them to retain their property





and lifestyles, so maintaining or strengthening their privileged position. In the event of demolition, on the other hand, this so-called favoured population loses its initial advantage (but is nevertheless compensated), which leads to a reduction (or a displacement) of EIs, in addition to the benefit of better security of the coast and its renaturation.

Is this *initial* inequality real? On the basis of the statistical databases available in France, it is difficult to validate this hypothesis of the favoured social position of coastal property owners (Long et al., 2019). Income levels are not known at the household level, so it is impossible to demonstrate a spatial correlation between high income and the presence of these households on the coast.

On the other hand, beyond the statistical demonstration, the Barnier Law nevertheless causes a reinforcement of the EIs.

In fact, the choice of the strategy of adaptation to coastal risks is linked to an economic calculation of the cost/benefit type. In the event of high land prices, the costs of demolition, including expropriation and compensation for owners, are such that the building of hard structures of defence will be preferred for financial reasons. Thus, in the town of Charron where the average price of property per sq.m is on average 1,800 euros, 180 houses were demolished following Storm Xynthia, while on the Ile de Ré where the price of land varies between 5,000 and 7,300 euros/sq.m on average, a seawall

construction plan was preferred for the island (source: www.meilleursagents.com). And therefore, the richer the owners (generally the case when the land prices are high), the more likely they are to be protected by a seawall instead of being subjected to managed retreat. In this case, for the same cost incurred, the mode of protection benefits them to a large degree, including in the long term (left-hand side of figure 4). In the event of low land prices, demolition is less expensive than the construction of a coastal defence, and will therefore probably be chosen, to the detriment of the owners, certainly

less wealthy, who will be subjected to managed retreat (right-hand side of figure 4). **There is therefore a reinforcement of environmental inequalities and, in particular, inequality in the treatment of territories, often suffered by the populations.**

However, it should be noted that over the long term, these conclusions can be qualified. We have in fact taken as a hypothesis an unchanged policy whatever the evolution of the hazard. This being largely uncertain, nothing indicates that

the current protection measures will be sufficient and if, therefore, adaptation and financing policies will not have to be reviewed. It would thus seem that, still in the long term, managed retreat leads the territories to be more resilient than when the choice to hold the coastal line is made; obviously, a new state of equilibrium has been reached in the system, but it appears to be more precarious and more fragile in the long term. The coastline is a system in perpetual evolution; we can then assume that this strategy will be costly and unsustainable in the long term and that a new trajectory will have

to be adopted.

In a context of climate change, with an increase in extreme events and rising sea levels, the French insurance system could be called into question fairly soon, not to mention the functioning of the Barnier Fund. Indeed, although 200,000 euros/year of receipts currently permit compensation for the people and property affected, what will happen in the future if


disasters show a tendency to increase in number and intensity? Should the percentage deducted from the CATNAT sur-
charge be raised? Will the insurance policyholders always agree to pay for the inhabitants of coastal areas? Should we
fear a system where insurers will no longer insure properties and people in risk areas, or where only the richest will be
able to pay very high premiums to continue living in these territories? Will the coast continue to be attractive? A study
conducted in South Florida by Theurer et al. (2018) for example, has shown following surveys that, faced with the increase
in sea level, it is the youngest owners (under 45) who are most inclined to move fairly early on, but on the contrary middle-
income earners are the least willing to move. However, along our coasts, the population tends to be ageing.

The choices between these different strategies for adapting to coastal risks are not simple and another parameter must be
taken into account, that of path dependency (Lawrence et al., 2018). The territories each have their own history and some
have historically chosen to gradually claw back from the sea land that has become rich and fertile for agriculture (polders).
Today, this land is found below sea level, protected by a seawall. How then is it possible to go back in time and have
people accept that this land be returned to the sea and serve as a buffer zone, welcoming sea water during submersions to
prevent inland flooding? The question of social and environmental justice arises here and could be considered as a crite-
rion as important as the economic one for the future choices to be made in terms of adaptation of coastal territories.

**Appendices**

**Appendix A:**

**Inequality in access to land:** This inequality represents accessibility to land, that is to say the possibility that a household
has to buy or rent property throughout its living area. Depending on certain factors, such as in our case, the proximity of
an attractive coastline, prices vary greatly, with a decrease in the price per sq.m from the coast to the interior. Certain
parts of the territory, therefore close to the coast, are inaccessible in residential terms for certain social categories.

**Inequality in exposure to coastal risks:** Only exposures to hazards from the sea such as submersion, erosion or rising
sea levels are taken into account. Exposure to these risks is uneven in the territory and depends on the morphology of the
coast (beach/cliff, silt/sand/pebbles, etc.) and on its evolution over time. Some sites may become exposed to hazards over
time for natural or man-made reasons.

**Inequality in access to the coast perceived as an amenity**: The coastline is understood here as a natural amenity, that
is to say as "local attributes that provide a set of benefits to people (especially climatic, aesthetic and recreational benefits),
[...] a contribution to the overall well-being or quality of life of the residents in a location, [...] as local characteristics
generating attractiveness." (Shaeffer and Dissart, 2018). It is recognised as a common good of humanity but sometimes
because of private developments on the coast, its access is no longer possible. We are also witnessing the privatisation of
portions of the coast for economic activities, access to which will be reserved for people paying an admission fee.



**Social inequality:** in our study, social inequality is only measured through the level of household income.

**Appendix B:**

| Indicators | Definition |
|---|---|
| Economic and property values | Refers to the economic value of a property |
| Accessibility to the coast | Represents the degree of physical accessibility of the coastline, made possible or otherwise by various public access points for pedestrians or vehicles |
| Environmental evolution of the coast | Defines how the geomorphology of the coastline will evolve over time |
| Natural hazard exposure | This concerns the exposure of people and property to coastal risks such as marine submersion, erosion, rising sea levels |
| Inhabitant feeling | Defines people's feelings and emotions |
| Sense of place | Translates the relationship, attachment to the place of the population |
| Social cohesion | Defines the links between people in a community, a group |

**Table B1: definition of indicators**

**Author contribution**

NL: Conceptualization, methodology, writing

PC: Conceptualization, methodology, visualization, writing

VK: Conceptualization, methodology, writing

**Declaration of Competing Interest**

The authors declare that they have no known competing financial interests or personal relationships that could have appeared to influence the work reported in this paper.

**Acknowledgements**

This work was supported by the Fondation de France.



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
