# Peer review of "Strategies for adapting to hazards and environmental inequalities in coastal urban areas: what kind of resilience for these territories?"

_Natural Hazards and Earth System Sciences, 2020_

## Referee Comment (RC1) · Anonymous Referee #1 · 5 Dec 2020

The topic is interesting and the article is well written. In paragraph 2.1 some data on the expected climate projections could be included to better present the case study. The figures could be enhanced.

---

## Referee Comment (RC2) · Anonymous Referee #2 · 18 Dec 2020

L'article est bien rédigé et n'a pas été publié à notre connaissance. Les références bibliographiques sont à jour, mais certaines mériteraient toutefois d'être complétées sur les aspects d'adaptation (par ex., mais il y pourrait y en avoir d'autres, le récent ouvrage Berdoulay et Soubeyran, 2020) et sur les indicateurs de résilience proposés par la communauté scientifique (ce qui permettrait aux auteurs de mieux positionner leurs choix d'indicateurs proposés en Annexe B).

L'introduction est très pertinente, elle ouvre véritablement la réflexion sur la résilience, l'adaptation, les inégalités environnementales. On peut regretter de perdre un peu ces aspects au fil de l'article, c'est dommage car c'est très intéressant. Pour éviter cela, les

auteurs pourraient davantage re-questionner ces aspects tant à partir de l'approche originale proposée (3.2. Method) qu'à partir des Âń Résultats Âż présentés dans la partie 4 de l'article. A ce titre, la méthodologie mériterait d'être davantage présentée et expliquée pour i) pouvoir la valider et identifier si elle peut facilement être transposable à d'autres territoires littoraux en France, en d'autres termes répondre à une des préconisations de l'article : produire une méthodologie générique ; ii) comprendre comment les auteurs obtiennent les résultats (partie 4). En l'état, on a parfois du mal à les vérifier.

Quelques pistes pour présenter la méthodologie :

- Les indicateurs pourraient être définis dans la partie Méthodologie de l'article et non en Annexe, et être appuyés par des sources. Les critères quantitatifs ou qualitatifs permettant d'évaluer ces indicateurs doivent être clairement énoncés, de même que leurs unités de mesures. - Pour chacun de ces critères, il faudrait expliquer comment se fait l'évaluation et à partir de quel(s) corpus : par exemple comment évaluer vous l'indicateur Âń Inhabitant feeling Âż ? Avez-vous réalisé des enquêtes, des entretiens ? sur quel échantillon ? quand avez-vous réalisés ces enquêtes ou entretiens etc. ? pour répondre à quelle question précise ? avez-vous interrogé les propriétaires sur leur niveau d'acceptation de démolition de leur propriété à CT et LT (en réponse à la partie 3.2.) ? De même, si je prends l'exemple de l'exposition à l'aléa et de l'inégalité à cette exposition : faites-vous référence à une inégalité perçue ? à une inégalité qui découlerait de l'analyse de la cartographie réglementaire établie dans les PPR ou les PPRL ? ou de cartographie réalisée dans le projet INEGALITTO etc.

Ces Âń questionnements Âż font que lorsque vous présentez vos résultats, on a un peu de mal à comprendre comment vous les avez obtenus, c'est dommage. Par exemple, pour les figures 2 et 3 : avez-vous analysé des outils réglementaires et des articles de loi (dans l'affirmative, des extraits de loi pourrait venir illustrer vos propos) pour proposer ces analyses ? Idem pour les tableaux 3 et 4 : à partir de quels corpus de données obtenez vous ces résultats ? avez-vous réalisé des enquêtes auprès des 3

groupes de populations ? etc.

L'article est intéressant mais il nécessite quelques approfondissements. Ces limites sont certainement dues au fait que les résultats s'appuient sur le projet de recherche INEGALITO et qu'il est toujours difficile de Âń prendre de la distance Âż entre un rapport de recherche, qui peut faire 100 pages, et un article qui doit rendre intelligible des années de recherche.

---

## Author Comment (AC1) · 29 Jan 2021

**Strategies for adapting to hazards and environmental inequalities in coastal urban areas: what kind of resilience for these territories?**

Nathalie Long, Pierre Cornut, Virginia Kolb

| Comments of anonymous Referee #1 | Answers |
|---|---|
|  | Thank you very much for your comments of our research paper. We respond below: |
| Some data on the expected climate projections could be included to better present the case study | We propose to complete the initial presentation of the case-study area with climate foresights (sea level rise, extreme events…) and additional bibliographic references about climate change on this region. |
| The figures could be enhanced | We suggest to add this layer to figure 1: https://coastal.climatecentral.org/map/9/-0.187/45.7858/?theme=sea_level_rise&map_type=coastal_dem_comparison&basemap=roadmap&contiguous=true&elevation_model=coastal_dem&forecast_year=2050&pathway=rcp85&percentile=p95&refresh=true&return_level=return_level_1&slr_model=kopp_2017 This layer represents the surface projected to be below annual flood level in 2050. For the other figures, the referee#1's comment is a bit too vague to ameliorate. |

---

## Author Comment (AC2) · 29 Jan 2021

**Strategies for adapting to hazards and environmental inequalities in coastal urban areas: what kind of resilience for these territories?**

Nathalie Long, Pierre Cornut, Virginia Kolb

| Comments of anonymous Referee #2 | Answers |
|---|---|
|  | Thank you very much for your detailed and thorough analysis of our research paper. Those comments are very helpful for revising and improving our paper, as well as the important guiding significance to our researches. We have studied comments carefully and we respond below point by point to the comments: |
| L'article est bien rédigé et n'a pas été publié à notre connaissance. Les références bibliographiques sont à jour, mais certaines mériteraient toutefois d'être complétées sur les aspects d'adaptation (par ex., mais il y pourrait y en avoir d'autres, le récent ouvrage Berdoulay et Soubeyran, 2020) et sur les indicateurs de résilience proposés par la communauté scientifique (ce qui permettrait aux auteurs de mieux positionner leurs choix d'indicateurs proposés en Annexe B). | Bibliography will be completed on coping strategies and resilience indicators. Like NHESS is an international review, we will use mainly international references. |
| L'introduction est très pertinente, elle ouvre véritablement la réflexion sur la résilience, l'adaptation, les inégalités environnementales. On peut regretter de perdre un peu ces aspects au fil de l'article, c'est dommage car c'est très intéressant. Pour éviter cela, les auteurs pourraient davantage re-questionner ces aspects tant à partir de l'approche originale proposée (3.2. Method) qu'à partir des Résultats présentés dans la partie 4 de l'article. | It is true that we probably did not insist enough on the resilience impacts in our discussion.
We suggest:
1) to add this short paragraph just after table 5, line 306:
"In terms of resilience, the results are more clear-cut: either in environmental or social matters, demolition is surely more resilient in the long terms. First, seawalls are only a short-term strategy given the century timescale of the sea level rise. The coastline is fixed and has no possibility of natural evolution, of coping to global change, which is contrary to the definition of a resilient system. Second, social cohesion in the case of demolition will be renewed in the long term, especially with the past adaptation strategy in the mind of local population. It is then essential to propose the adaptation strategy that will best preserve social cohesion, because it is the guarantor of |

| | spontaneous solidarity following the occurrence of an extreme event. This solidarity makes it possible to define this society as more resilient because it is able to better cope and adapt." |
|---|---|
| | 2) To add this sentence at the end of paragraph line 336:
"Following this argument, the conclusion is totally opposite in resilience matter: poorer territories should become more resilient than richer ones in the long term. " |
| A ce titre, la méthodologie mériterait d'être davantage présentée et expliquée pour i) pouvoir la valider et identifier si elle peut facilement être transposable à d'autres territoires littoraux en France, en d'autres termes répondre à une des préconisations de l'article : produire une méthodologie générique ; ii) comprendre comment les auteurs obtiennent les résultats (partie 4). En l'état, on a parfois du mal à les vérifier. | Our methodology is not proved to be transposable, at this stage of our research at least. We based it on our experience in the Inegalitto project i.e. on La Rochelle and Saint-Brieuc coasts. We have some confidence in the transposability of the method, however, since:
- We cross qualitative and qualitative indicators
- We take into account the coast typology and the actor's behaviours, acknowledging the fact that 'space matters'
And we pointed in the conclusion, line 355, that path dependency is of high importance to understand the coastal strategies.
To address this referee's comment, we suggest to add these sentences in line 356, just after the Lawrence reference:
"Our methodology, based on qualitative and quantitative indicators, has been inspired by our research experience in the Inegalitto project. Although we took into account the coast typology and the actor's behaviours, acknowledging the fact that 'space matters', path dependency is still questionable. We this intend in coming months to test our methodology on other coastal area, first in the same juridic French context, second in other countries. " |
| Quelques pistes pour présenter la méthodologie :
- Les indicateurs pourraient être définis dans la partie Méthodologie de l'article et non en Annexe, et être appuyés par des sources. | OK, this will be done in the final version (in appendix, we suggest, in order to avoid a too long core text) |

| | |
|---|---|
| Les critères quantitatifs ou qualitatifs permettant d'évaluer ces indicateurs doivent être clairement énoncés, de même que leurs unités de mesures. - Pour chacun de ces critères, il faudrait expliquer comment se fait l'évaluation et à partir de quel(s) corpus : par exemple comment évaluer vous l'indicateur ' Inhabitant feeling'? Avez-vous réalisé des enquêtes, des entretiens? sur quel échantillon ? quand avez-vous réalisés ces enquêtes ou entretiens etc. ? pour répondre à quelle question précise ? avez-vous interrogé les propriétaires sur leur niveau d'acceptation de démolition de leur propriété à CT et LT (en réponse à la partie 3.2.)? | 1/ First, more details could be added in section *3.1 Results from the Inegalitto project* to describe the qualitative interview carried out during the project (main objectives of surveys, profile of people surveyed, date, etc).

 2/ Secondly, we explain the qualitative evaluation of the indicators in the text (lines 206 to 216). As we have 3 levels for each the indicator (improvement, neutral, degradation), and given our knowledge of the field (including qualitative interviews), the exercise is quite easy. For example, 'inhabitant feeling' is easily estimated as positive for PO and IPM in the case of seawall, and negative in the case of managed retreat: our interviews as well and media papers show it clearly.
 What we did practically is to discuss among us, for each of the 12 situations, whether the strategies produced +, 0 or – for the indicator, using data and interviews described in the paper.
 To address the Referee's comment, maybe could we add this in the text between lines 206 to 216.

 3/ The level of owner acceptance was not assessed during the surveys. This indicator is not used in our study (indicators list at lines 210 to 213). |
| De même, si je prends l'exemple de l'exposition à l'aléa et de l'inégalité à cette exposition : faites-vous référence à une inégalité perçue ? à une inégalité qui découlerait de l'analyse de la cartographie réglementaire établie dans les PPR ou les PPRL ? ou de cartographie réalisée dans le projet INEGALITTO etc. | Inequality in risk exposure is not a perceived inequality. It is measured by an indicator based on the profile of households and their distribution in space and on areas at risk of flooding, inundation and erosion (Long et al., 2019). We will add in the text line 161 a more detailed description of indicators developed in the Inegalitto project (section 3.1). To improve our definition of inequalities, this text could be add line 65
 "But, two approaches to inequality coexist: either the definition is based more on the point of view of the individual, considering that inequalities do not exist as such, but rather when they are felt by individuals. In this case, inequality is defined as "a difference that is perceived or experienced as unfair, as not ensuring the same opportunities for everyone" (Brunet et al., 1992); or it is considered that inequality arises when there is an unequal distribution of goods among individuals within society. In this case, inequality exists when an individual or a population holds resources, has access to certain goods or services, to certain practices unlike others. This definition is based on the existence of a hierarchical scale common to the whole of society and on |

| | which the vectors of inequality are uniformly classified. This second approach is used here." |
|---|---|
| Ces 'questionnements' font que lorsque vous présentez vos résultats, on a un peu de mal à comprendre comment vous les avez obtenus, c'est dommage. Par exemple, pour les figures 2 et 3 : avez-vous analysé des outils réglementaires et des articles de loi (dans l'affirmative, des extraits de loi pourrait venir illustrer vos propos) pour proposer ces analyses ? Idem pour les tableaux 3 et 4 : à partir de quels corpus de données obtenez vous ces résultats ? avez-vous réalisé des enquêtes auprès des 3 groupes de populations ? etc. | We hope that the clarifications made in the Methodology section will lead to a better understanding of our results. To obtain figures 2 and 3 and tables 3 and 4, based on the results obtained previously and on our discussions, we have tried to generalize these cause-effect relationships between adaptation strategies and inequalities. At this stage, we did not use any regulatory tools or articles of law. |
| L'article est intéressant mais il nécessite quelques approfondissements. Ces limites sont certainement dues au fait que les résultats s'appuient sur le projet de recherche INEGALITO et qu'il est toujours difficile de 'prendre de la distance' entre un rapport de recherche, qui peut faire 100 pages, et un article qui doit rendre intelligible des années de recherche. | Thanks again for your comments and suggestions; we hope to have responded to all of your remarks to make the results of our work more accessible and intelligible. |

---

## Author Response (AR1)

**Strategies for adapting to hazards and environmental inequalities in coastal urban areas: what kind of resilience for these territories?**

Nathalie Long, Pierre Cornut, Virginia Kolb

| Comments of anonymous Referee #1 | Answers |
|---|---|
|  | Thank you very much for your comments of our research paper. We respond below: |
| Some data on the expected climate projections could be included to better present the case study | We complete the initial presentation of the case-study area with climate foresights by additional bibliographic references about climate change on this region: Marcos et al., 2007 and Dodet et al., 2019, which calculated a sea-level rise of ~ 3mm/year. Other information about the characteristic of Xynthia storm was added to show the extreme intensity of this storm.
"This storm generated a storm surge reached its maximum in the centre of the Bay of Biscay, with a maximum of 1.5m (harbour of La Pallice , La Rochelle) in the same time as the high tide, resulting a total water level of 8.01m above marine chart datum in La Pallice (Bertin et al., 2012). Material damage was significant and lives were lost. In addition to this type of extreme hazard, there is a slower hazard to be taken into consideration: the sea level rise. As part of the global change, there is a ~3mm/year sea level rise in the Bay of Biscay (Marcos et al., 2007; Dodet et al., 2019) ." |
| The figures could be enhanced | A new image completes figure 1: The map represents the surface projected to be below annual flood level in 2050. |

| Comments of anonymous Referee #2 | Answers |
|---|---|
|  | Thank you very much for your detailed and thorough analysis of our research paper. Those comments are very helpful for revising and improving our paper, as well as the important guiding significance to our researches. We have studied comments carefully and we respond below point by point to the comments: |
| L'article est bien rédigé et n'a pas été publié à notre connaissance. Les références bibliographiques sont à jour, mais certaines mériteraient toutefois d'être complétées sur les aspects | Bibliography was completed on coping strategies: Nelson et al., 2007; Klein et al., 2003; Rocle et al., 2020. See our modifications in lines 58 to 63. |

| | |
|---|---|
| d'adaptation (par ex., mais il y pourrait y en avoir d'autres, le récent ouvrage Berdoulay et Soubeyran, 2020) et sur les indicateurs de résilience proposés par la communauté scientifique (ce qui permettrait aux auteurs de mieux positionner leurs choix d'indicateurs proposés en Annexe B). | The reference Assarkhaniki et al. 2020 is added (line 240) in order to better justify the interactions between inequality and resilience.

The proposed indicators are classified according to the main types of environmental inequalities: economic or social access inequalities (Indicator: Economic and property values), access to environmental amenities (Indicators: Accessibility to the coast, Environmental evolution of the coast); risk exposure inequalities (Indicator: Natural hazard exposure); and finally social and cultural inequalities (Indicators: Inhabitant feeling, Sense of place, Social cohesion). They allow assessing inequalities and not resilience directly. |
| L'introduction est très pertinente, elle ouvre véritablement la réflexion sur la résilience, l'adaptation, les inégalités environnementales. On peut regretter de perdre un peu ces aspects au fil de l'article, c'est dommage car c'est très intéressant. Pour éviter cela, les auteurs pourraient davantage re-questionner ces aspects tant à partir de l'approche originale proposée (3.2. Method) qu'à partir des Résultats présentés dans la partie 4 de l'article. | 1)   We added this short paragraph just after table 6 (lines 351-358) :
"In terms of resilience, the results are more clear-cut: either in environmental or social matters, demolition is surely more resilient in the long terms. First, seawalls are only a short-term strategy given the century timescale of the sea level rise. The coastline is fixed and has no possibility of natural evolution, of coping to global change, which is contrary to the definition of a resilient system. Second, social cohesion in the case of demolition will be renewed in the long term, especially with the past adaptation strategy in the mind of local population. It is then essential to propose the adaptation strategy that will best preserve social cohesion, because it is the guarantor of spontaneous solidarity following the occurrence of an extreme event. This solidarity makes it possible to define this society as more resilient because it is able to better cope and adapt."

2) We added this sentence at the end of paragraph line 380:
"Following this argument, the conclusion is however totally opposite in resilience matter: poorer territories should become more resilient than richer ones in the long term. " |
| A ce titre, la méthodologie mériterait d'être davantage présentée et expliquée pour i) pouvoir la valider et identifier si elle peut facilement être transposable à d'autres territoires littoraux en | Our methodology is not proved to be transposable, at this stage of our research at least. We based it on our experience in the Inegalitto project i.e. on La Rochelle and Saint-Brieuc coasts. We have some confidence in the transposability of the method, however, since: |

| | |
|---|---|
| France, en d'autres termes répondre à une des préconisations de l'article : produire une méthodologie générique ; ii) comprendre comment les auteurs obtiennent les résultats (partie 4). En l'état, on a parfois du mal à les vérifier. | - We cross qualitative and qualitative indicators
- We take into account the coast typology and the actor's behaviours, acknowledging the fact that 'space matters'
And we pointed in the conclusion, lines 400-404, that path dependency is of high importance to understand the coastal strategies.

To address this referee's comment, we added these sentences in line 401, just after the Lawrence reference:
"Our methodology, based on qualitative and quantitative indicators, has been inspired by our research experience in the Inegalitto project. Although we took into account the coast typology and the actor's behaviours, acknowledging the fact that 'space matters', path dependency is still questionable. We this intend in coming months to test our methodology on other coastal area, first in the same juridic French context, second in other countries. " |
| Quelques pistes pour présenter la méthodologie :
- Les indicateurs pourraient être définis dans la partie Méthodologie de l'article et non en Annexe, et être appuyés par des sources. | OK, this is done in the final version (new table 2, line 242). |

| Les critères quantitatifs ou qualitatifs permettant d'évaluer ces indicateurs doivent être clairement énoncés, de même que leurs unités de mesures. - Pour chacun de ces critères, il faudrait expliquer comment se fait l'évaluation et à partir de quel(s) corpus : par exemple comment évaluer vous l'indicateur ' Inhabitant feeling'? Avez-vous réalisé des enquêtes, des entretiens? sur quel échantillon ? quand avez-vous réalisés ces enquêtes ou entretiens etc. ? pour répondre à quelle question précise ? avez-vous interrogé les propriétaires sur leur niveau d'acceptation de démolition de leur propriété à CT et LT (en réponse à la partie 3.2.)? | 1/ First, more details were added in section *3.1 Results from the Inegalitto project* to describe the qualitative interview carried out during the project (main objectives of surveys, profile of people surveyed, date, etc). See lines 192-195 "The surveys were carried out in 2017 in Aytré and in 2018 in Charron. The people interviewed were local politicians, associations and residents. The aim surveys were to analyze residents' representations of coastal risk, to address the issue of compensation for households exposed to coastal risk and to compare differential treatment between areas."

2/ Second, we explained the qualitative evaluation of the indicators in the text (lines 230 to 236). As we have 3 levels for each the indicator (improvement, neutral, degradation), and given our knowledge of the field (including qualitative interviews), the exercise is quite easy. For example, 'inhabitant feeling' is easily estimated as positive for PO and IPM in the case of seawall, and negative in the case of managed retreat: our interviews as well and media papers show it clearly.
What we did practically is to discuss among us, for each of the 12 situations, whether the strategies produced +, 0 or – for the indicator, using data and interviews described in the paper.
To address the Referee's comment, we add this in the text between lines 244 to 248 (under table 2).
"These indicators are estimated for each of the 12 situations according to a simple qualitative scale: improvement or preservation (with a nuance depending on whether the populations are high concerned or low concerned), neutrality, degradation (with the same nuance). This level of assessment is given from our knowledge of the field and the interviews. For example, 'inhabitant feeling' is easily estimated as positive for PO and IPM in the case of seawall, and negative in the case of managed retreat: our interviews as well and media papers show it clearly. "

3/ The level of owner acceptance was not assessed during the surveys. This indicator is not used in our study (indicators list in table 2). |
| De même, si je prends l'exemple de l'exposition à l'aléa et de l'inégalité à cette exposition : faites-vous référence à une inégalité perçue ? à une inégalité qui découlerait de l'analyse de la | Inequality in risk exposure is not a perceived inequality. It is measured by an indicator based on the profile of households and their distribution in space and on areas at risk of flooding, inundation and erosion (Long et al., 2019). We added a few lines to give some |

| | |
|---|---|
| cartographie réglementaire établie dans les PPR ou les PPRL ? ou de cartographie réalisée dans le projet INEGALITTO etc. | details but an extensive description of the *index* developed in the Inegalitto project is not the issue of this article. We used the term "index" to describe the results of the Inegalitto project and avoid the confusion with the indicators used to assess the impact of adaptation policy on inequalities. See lines 180 to 186:
"The first part of this project consisted in mapping environmental inequalities using *indexes* to *measure* inequalities in access to natural and anthropogenic amenities, inequalities in exposure to natural and industrial risks, and inequalities on the economic level *from several databases. As example, index of inequalities in exposure to risk is based on distribution households in space and on areas at risk of flooding, inundation and erosion (more details* in *Long et al., 2019)*. These inequalities were then compared with social ones, defined by socio-demographic data at the household level, from national statistical databases."

To improve our definition of inequalities, we added this text line 69:
"Two approaches to inequality coexist: either the definition is based on individuals' point of view, considering that inequalities do not exist as such but rather that they are felt by individuals. In this case, inequality is defined as "a difference that is perceived or experienced as unfair, as not ensuring the same opportunities for everyone" (Brunet et al., 1992). The second approach considers that inequality arises when there is an unequal distribution of goods among individuals within society. In this case, inequality exists when an individual or a population holds resources, has access to certain goods or services and to certain practices, unlike others. This definition is based on the existence of a hierarchical scale common to the whole of society and on which the vectors of inequality are uniformly classified. This second approach is used here." |
| Ces 'questionnements' font que lorsque vous présentez vos résultats, on a un peu de mal à comprendre comment vous les avez obtenus, c'est dommage. Par exemple, pour les figures 2 et 3 : avez-vous analysé des outils réglementaires et des articles de loi (dans l'affirmative, des extraits de loi pourrait venir illustrer vos propos) pour proposer ces analyses ? Idem pour les tableaux 3 et 4 : à partir de quels corpus de données obtenez vous ces | We hope that the clarifications made in the Methodology section will lead to a better understanding of our results. To obtain figures 2 and 3 and tables 3 and 4, based on the results obtained previously and on our discussions, we have tried to generalize these cause-effect relationships between adaptation strategies and inequalities. At this stage, we did not use any regulatory tools or articles of law. |

| | |
|---|---|
| résultats ? avez-vous réalisé des enquêtes auprès des 3 groupes de populations ? etc. | |
| L'article est intéressant mais il nécessite quelques approfondissements. Ces limites sont certainement dues au fait que les résultats s'appuient sur le projet de recherche INEGALITO et qu'il est toujours difficile de 'prendre de la distance' entre un rapport de recherche, qui peut faire 100 pages, et un article qui doit rendre intelligible des années de recherche. | Thanks again for your comments and suggestions; we hope to have responded to all of your remarks to make the results of our work more accessible and intelligible. |